# Antioxidant Capacity and NF-kB-Mediated Anti-Inflammatory Activity of Six Red Uruguayan Grape Pomaces

**DOI:** 10.3390/molecules28093909

**Published:** 2023-05-05

**Authors:** Emiliana Fariña, Hellen Daghero, Mariela Bollati-Fogolín, Eduardo Boido, Jorge Cantero, Mauricio Moncada-Basualto, Claudio Olea-Azar, Fabio Polticelli, Margot Paulino

**Affiliations:** 1Área Bioinformática, DETEMA, Facultad de Química, Universidad de la República, Gral. Flores 2124, C.P. 11800, C.C. 1157, Montevideo 11600, Uruguay; 2Cell Biology Unit, Institut Pasteur Montevideo, Mataojo 2020, Montevideo 11400, Uruguay; 3Laboratorio de Enología, DETEMA, Facultad de Química, Universidad de la República, Gral. Flores 2124, C.P. 11800, C.C. 1157, Montevideo 11600, Uruguay; 4Medical Research Center, Facultad de Ciencias de la Salud, Universidad Nacional del Este, Minga Guazú 7420, Paraguay; 5Programa Institucional de Fomento a la Investigación, Desarrollo e Innovación, Universidad Tecnológica Metropolitana, San Joaquín 8940577, Chile; 6Facultad de Cs. Químicas y Farmacéuticas, Universidad de Chile, Dr. Carlos Lorca Tobar 964, Región Metropolitana, Santiago de Chile 8380494, Chile; 7Department of Sciences, University Roma Tre, Viale G. Marconi 446, 00146 Rome, Italy; 8National Institute of Nuclear Physics, Roma Tre Section, Via della Vasca Navale 84, 00146 Rome, Italy

**Keywords:** grape pomace, phenols, antioxidants, DPPH^·^, ABTS^·+^, ESR, ORAC, FRAP, NF-κB, molecular docking

## Abstract

Grape pomaces have a wide and diverse antioxidant phenolics composition. Six Uruguayan red grape pomaces were evaluated in their phenolics composition, antioxidant capacity, and anti-inflammatory properties. Not only radical scavenging methods as DPPH^·^ and ABTS^·+^ were employed but also ORAC and FRAP analyses were applied to assess the antioxidant potency of the extracts. The antioxidant reactivity of all extracts against hydroxyl radicals was assessed with ESR. The phenol profile of the most bioactive extract was analyzed by HPLC-MS, and a set of 57 structures were determined. To investigate the potential anti-inflammatory activity of the extracts, Nuclear Factor kappa-B (NF-κB) modulation was evaluated in the human colon cancer reporter cell line (HT-29-NF-κB-hrGFP). Our results suggest that Tannat grapes pomaces have higher phenolic content and antioxidant capacity compared to Cabernet Franc. These extracts inhibited TNF-alpha mediated NF-κB activation and IL-8 production when added to reporter cells. A molecular docking study was carried out to rationalize the experimental results allowing us to propose the proactive interaction between the NF-κB, the grape extracts phenols, and their putative anti-inflammatory bioactivity. The present findings show that red grape pomace constitutes a sustainable source of phenolic compounds, which may be valuable for pharmaceutical, cosmetic, and food industry applications.

## 1. Introduction

Reactive oxygen species (ROS) are products of normal cellular metabolism and are well recognized as both beneficial and deleterious species [1,2]. A balance between the formation of ROS and their elimination is essential for normal cellular function. When such balance is disrupted as a result of excessive generation or defective removal of these species, cells are damaged due to oxidative stress. At high concentrations, ROS can mediate severe damage to cell structures, including nucleic acids, lipids, and proteins [1,3], and to cellular functions [4].

Increasing experimental and clinical evidence show the involvement of oxidative stress induced by ROS and Nitrogen Oxidative Species (NOS) in a variety of disorders, including cancer [1,5], atherosclerosis [6,7], neurodegenerative diseases, and aging [5,8]. This suggests that agents with the ability to protect against ROS may be therapeutically useful in the prevention and treatment of these disorders.

Many antioxidant compounds, such as polyphenols, are found in several fruits and vegetables. Grapes and wines, which can contain high levels of phenolic antioxidants, have been shown to exert beneficial effects on health.

Particularly, these compounds have been demonstrated to possess many biological properties as antioxidants [9,10], modulators of the inflammatory cascade [9], inhibitors of LDL-c oxidation [11], and antiplatelets [12].

Grape pomace is the main solid organic waste from wine industry. It is the material resulting from the pressing of the grapes, and it consists of skin, pulp, seeds, and stems [13]. In Uruguay, the National Institute of Viticulture (INAVI) [14] publishes statistics every year. The publications of the Annual Statistics provide information regarding the Harvest, in which the historical variables considered are: Declarations Processed, Grape Varieties, by Type and Typification—expressed in Plants, Surface, Production, Productivity and Density of Plantation—and Destination in kilograms. The 2022 report indicated that a Total National Production of 106,672,752 kg of grapes was recorded, of which 10% was produced as grape marc, i.e., 10,667,275 kilos. Thus, grape pomace potentially constitutes a very abundant and relatively inexpensive source of a wide range of polyphenols [13,14,15]. Due to their diverse biological activity, mainly as antioxidants and anti-inflammatory compounds, there is a vast array of potential applications for grape pomace extractable components as ingredients of functional foods and animal feeds, cosmetics, and pharmaceuticals [16,17,18,19].

In this work, we extracted and analyzed by HPLC-MS the phenolic content of Uruguayan red grape pomaces derived from five Tannat samples and one from Cabernet Franc. The extracts were further characterized in terms of their antioxidant and scavenging activity by using different methods: Oxygen radical antioxidant capacity-fluorescein (ORAC), Ferric Reducing Antioxidant Power (FRAP), Electron Spin Resonance (ESR), 2,2-diphenyl-picrylhydrazyl (DPPH^·^), and 2,2′-azino-bis(3-ethylbenzothiazoline-6-sulphonic acid) (ABTS^·+^).

Furthermore, we tested whether those samples were able to modulate NF-κB activity in vitro using the human colon cancer reporter cell line HT-29-NF-κB-hrGFP. Finally, a molecular docking between the identified phenolic compounds and NF-κB crystallographic structure (PDB id 1VKX) allowed to uncover the molecular basis of the interaction and therefore provided a likely explanation for the observed NF-κB modulation.

## 2. Results and Discussion

### 2.1. Development of the Extraction Method

In order to select the best mixture that yields high polyphenols content, different solvent mixtures and sample weights were employed. For the first step and aiming at finding the ethanol/water proportion that extracted the most polyphenols, 1 g of dry pomace samples were submitted to maceration and extraction with different ethanol/water mixtures (100/0, 80/20, 50/50, 40/60, 20/80, and 0/100). Results are shown in Figure 1, where the total phenolic content is expressed in mg GAE/L (GAE, Gallic Acid Equivalent).

The highest concentration of total polyphenols was obtained when a 50:50 ethanol:water mixture was used. These results were in accordance with the polarity of the phenols expected to be extracted since in grape pomace, it is very common to find glycosylated hydrophilic polyphenols. The increase in solvent polarity is associated with an increase in the detected equivalents measured as mg GAE/L. After reaching the optimum total polyphenol values in the 50:50 ethanol:water mixture, the amount of phenols extracted decreases due to the higher water content, which does not extract polyphenols with a moderate polarity. On the other hand, the extract derived from Tannat samples showed higher polyphenols content than that derived from Cabernet Franc.

### 2.2. Antioxidant Capacity Evaluation

Once the extraction method was optimized, the antioxidant capacity was evaluated for the six samples of grape pomace using two sample weights (5 g and 10 g).

There are many assays available for estimating antioxidant capacity, and, depending on the chemical reactions involved, they can be roughly divided into hydrogen atom transfer- (HAT) or electron transfer- (ET) based assays [20]. In HAT-based assays, the antioxidant transfers a hydrogen atom to the thermally generated oxidant radical species. An example of this type of assay is the ORAC assay, in which there is a hydrogen transfer to the peroxyl radical generated. ET-based assays measure the reducing capacity of an antioxidant. Some techniques based on this mechanism are the FC, the FRAP, the DPPH, and the ABTS methods [20].

In this work, different methods associated with different antioxidants mechanisms were used, and the results for 10 g dry weight samples are shown in Figure 2 (see Appendix A for the results for 5 and 10 g). Since the total phenolic contents and the antioxidant capacity were similar between 5 g and 10 g dry weight samples, the extraction obtained from 10 g dry weight was used for further analysis.

#### 2.2.1. Free Radical Scavenging Capacity Measured by DPPH

The results were expressed in mg eq AAS/L, being AAS ascorbic acid. The Tannat extracts showed the best scavenging capacity with DPPH radical, with Tannat B displaying the highest scavenging percentage (Appendix A). Since both methods are based on the reducing capacity of the polyphenolic compounds of the extracts, a good correlation was expected between DPPH and FC method. DPPH assay displayed a positive correlation with the total polyphenolic content (Figure 2a). A relationship between both DPPH and FC emerges with an R^2^ of 0.92.

#### 2.2.2. ABTS^+^ Assays

The scavenging activity of the ABTS radical was determined spectrophotometrically. The Tannat 1 and Tannat B extracts displayed the best radical scavenging activity, while the Cabernet Franc extract had the lowest one (Appendix A). These results were similar to the DPPH scavenging activity. Additionally, the correlation obtained between total polyphenols content and the scavenging activity of the ABTS radical (expressed in mg eq/L) was similar to that of DPPH (Figure 2a,b).

When comparing DPPH^·^ and ABTS^·+^ methods, both radicals exhibited excellent stability under experimental conditions; their correlation being R^2^ = 0.86 (Figure 2e), associated with the differences in the antioxidant mechanisms used to detect the antioxidant capacity. Indeed, the two methods have different mechanisms for the radical generation: DPPH^·^ is a free radical that can be obtained directly without preparation, while the ABTS^·^ has to be generated after a chemical reaction. Another difference is that ABTS^·+^ is useful to measure the activity of hydrophilic and lipophilic compounds, while the DPPH^·^ can only be dissolved in organic solvents. ABTS^·+^ radical has also the advantage that its spectrum displays maximum absorbance values at 414, 654, 754, and 815 nm in an alcoholic medium, while DPPH^·^ displays a single absorbance peak at 515 nm.

#### 2.2.3. FRAP Assays

This methodology implies a redox reaction between the reducing compounds and an antioxidant agent (the ferric salt) that, when reduced, produces a blue color through a reaction of tripiridyltriazine. The values obtained for the extracts, shown in Appendix A, display a strong correlation of FRAP measurement with the total polyphenolic content (R^2^ = 0.97; Figure 2d). It is worthwhile to mention that there are several groups able to transfer electrons in the ferric salt [Fe (III) (TPTZ) 2] + 3, and, for this reason, the FRAP value will be increased.

#### 2.2.4. ORAC-FL Assay

In these experiments, a kinetic profile of fluorescein consumption induced by the peroxyl radicals generated through the ABAP thermolysis was obtained in the presence of extracts. The results were obtained by the interpolation of the area under the curve for the attenuation of the signal and that of the Trolox calibration plot. The results were expressed as millimolar Trolox equivalents (mMTE).

The results showed that the ORAC values differ very little between the six extracts, but the Tannat B sample showed the highest ability to inhibit the signal of fluorescein oxidation (Appendix A). In this case, the correlation between ORAC and total polyphenolic content was only 41% (Figure 2d).

The chemical structure of polyphenolic compounds determines the extraction composition, i.e., the glycosylated polyphenols will be extracted mainly in most polar extraction mixtures. On the other hand, another important difference to consider is that in the ORAC assay, the antioxidant is evaluated by its capacity to donate a hydrogen atom to the radical centered on the oxygen (HAT mechanism), while in the FRAP method, the reducing power is measured by an electron transfer mechanism (ET). Moreover, in ORAC-FL assay, antioxidants may display not only the ability to trap peroxyl radicals but also alkoxyl radicals with a very different kinetics that might contribute to some difference in the values [21]. Taking into account these differences between mechanisms, the correlation between the two results, with an R^2^ of 0.70, was considered satisfactory (Figure 2f).

### 2.3. Antioxidant Reactivity by ESR Assay

In ESR spectra, a hyperfine pattern of four lines was observed corresponding to the spin adduct DMPO-OH (major) and a pattern of six lines of lower intensity corresponding to the spin adduct DMPO-CH3 generated by the used solvent mixture (Figure 3). In Appendix A, the results are shown as percentage of hydroxyl radical scavenging.

A decreased signal intensity of the spin adducts DMPO-OH was observed for all tested samples (available upon request to the authors). In some cases, an increase in the signal strength was attributable to the trapping of carbon radical species that are the products of the reaction of the hydroxyl radical with the polyphenols of the samples. Figure 3 shows the ESR spectra corresponding to Tannat B extract since it was the sample that displayed the best scavenging activity of the generated hydroxyl radical. The decrease in the signal of the adduct DMPO-OH is shown in a dark blue dotted box, and the increase in the adduct signal DMPO-CR is shown in a green dotted box. The assignment of these signals was made from the registration of the ESR spectra of adducts corresponding to the trapping of the hydroxyl radical and of carbon-centered radicals, and Figure 3C,D, respectively.

The Spin Trapping technique allowed to study the hydroxyl radical through the increase in its mean lifetime by the formation of a spin adduct with the DMPO trapping. This is a difference with respect to the ORAC technique in which an indirect observation of the oxygen radical reactivity is made, postulating that a peroxyl radical or, alternatively an alcohoxyl radical is mainly formed (black lines in Figure 3B,D).

Based on the results described above, the electronic spin resonance, in conjunction with the technique of spin entrapment, appears to be a useful tool for the determination of the antioxidant activity of natural products. In addition, information was obtained on the formation of new radical species centered on carbon, possibly derived from polyphenolic compounds.

Therefore, it can be concluded that the solvent mixture used in the extraction allows to obtain phenolic compounds with antioxidant capacity.

### 2.4. HPLC-MS

In Table 1, the identified phenolic compounds from the most bioactive grape extract are listed.

We have identified four main phenol scaffolds in the mixture: phenolic acids, flavonols, anthocyanins, and flavan-3-ols. Each of these compound classes has been shown to possess multiple bioactive properties, which makes this naturally occurring mixture highly intriguing. Our research has provided evidence of the quality of this profile for the first time. Flavonols, especially quercetin, are widely used due to their proven antioxidant, anti-inflammatory, and neuroprotective effects [22,23]. Similarly, flavan-3-ols are commonly used as scaffolds because of their antioxidant capacity, as seen in green tea [24]. Overall, the complexity and diversity of the structures that can result from the presence of polymerization products in the mixture of polyphenols underscore the importance of studying the composition and properties of these mixtures in relation to their potential biological activities.

A total of 43 compounds were unambiguously identified, including 7 flavonols, 15 anthocyans, 5 phenolic acids, and 16 flavanols. The flavan-3-ols set includes the procyanidin trimer C2 (Table 1, row 21) and four sets of procyanidin trimers (Table 1, rows 25, 26, 31, and 32). In these last cases, given that all procyanidin trimers have the same molecular weight, we cannot assess which one is present in the mixture. All possible trimeric combinations of catechin (C) and epicatechin (E) monomers (CCC, ECC, CEC, CCE, EEC, ECE, CEE, and EEE) and the linkage between them C4-C8 (48) or C4-C6 (46), that generates four putative linkage classes (4848, 4846, 4648, and 4646), accounts for thirty-two modelled trimeric structures.

Different studies have investigated the anti-inflammatory effects of the phenolic compounds of grape pomaces. Pistol and coworkers [22] explored the use of pomace to reduce inflammation in Caco-2 intestinal cells. They found that the polyphenol extract was effective in down-regulating multiple inflammation signaling pathways, including NF-κB. Similarly, Calabriso and coworkers [23] investigated the effects of grape pomace from *Vitis vinifera* cv Negramaro on inflammation in intestinal and endothelial cells. They found that treatment with the pomace-derived polyphenols resulted in down-regulation of NF-κB pro-inflammatory signaling pathways and pro-inflammatory cytokines and chemokines in both cell types. Overall, both studies suggest that the anti-inflammatory effects of polyphenols may be attributed, at least in part, to its ability to inhibit NF-κB activation. Based on these findings, we were interested in analyzing the effect of these extracts on inflammation evaluating NF-κB activation and IL-8 production [22,23].

All in all, a database of seventy compounds was created for the *in silico* procedures that allowed us to rationalize the putative relationship between the composition of our phenols mixture and their capacity to interfere in the NF-κB-DNA complex (see Section 2.7).

### 2.5. Modulation of NF-κB Activity In Vitro

A large body of evidence demonstrates that certain phytochemicals, such as polyphenols, can attenuate inflammatory processes by blocking the NF-κB pathway [25]. Many reports demonstrated that polyphenols can modulate the inflammatory response both in vitro and in vivo [26]. In particular, there are reports of polyphenols from grape seeds and pomace extracts with anti-inflammatory activities in vitro [16,25]. Thus, we next evaluated the ability of the extracts to modulate NF-κB activity in a model of epithelial intestinal reporter cells, HT-29-NF-κB-hrGFP. These cells were stably transfected with a plasmid containing six NF-κB responsive elements, driving the expression of the GFP [27]. The relevance of this model relies on the ability to directly measure transcriptional activity of the NF-κB pathway, the possibility of using live cells and the convenience of detecting GFP expression by flow cytometry. HT-29-NF-κB-hrGFP cells were incubated with pro-inflammatory stimulus (10 ng/mL of TNF-α) and the different pomace extracts, and finally, the NF-κB activation was measured by GFP expression measured by flow cytometry. As shown in Figure 4A, only Tannat B extract was able to significantly inhibit TNF-α mediated NF-κB activation. When comparing the antioxidant activity of the extracts with the different assays, Cabernet Franc and Tannat 4 were the samples with the least antioxidant activity, and those extracts were not able to modulate NF-κB activation levels (similar to the untreated control). Further studies are needed to assess if the extracts can modulate the inflammatory response *in vivo*.

### 2.6. IL-8 Quantification in HT-29-NF-κB-hrGFP Culture Supernatants

In addition to the NF-κB activation assay, we next analyzed IL-8 production. This cytokine is secreted by epithelial cells, and it is an important mediator of the immune reaction in the innate immune system response. It is a pro-inflammatory cytokine and a target gene of the NF-κB transcription factor [28]. Thus, IL-8 production was quantified in HT-29-NF-κB-hrGFP cell supernatants. Cells were incubated with the different tannat extracts and TNF-α for 24 h. After that, supernatants were collected, and IL-8 levels were quantified by an ELISA. Results are shown in Figure 4B. Except for cells treated with Tannat 4, all tannat extracts (1, B, 2 and 3) showed a decrease in the IL-8 production compared to the control cells (treated only with TNF-α). These results indicate that Tannat B extract was capable of inhibiting IL-8 production from intestinal epithelial cells through modulation of NF-κB activation, while tannat 1, 2, and 3 extracts blocked IL-8 through a different mechanism than NF-κB signaling pathway.

### 2.7. Molecular Docking of NF-κB-Phenols

The program Site Finder (part of the Molecular Operating Environment (MOE) package. Montréal, QC, Canada, H3A 2R7)) was used to predict putative binding sites in the NF-κB structure. This procedure detected three main regions: two main contact sites outside the DNA-protein interface and a third site located at the interface, where we suppose the main interference with protein binding to DNA should occur. The result is shown in Figure 5.

Subsequently, a blind docking (with the search space encompassing the whole NF-κB-DNA complex) was performed using Autodock Vina and the phenolic compounds identified by HPLC-MS. These simulations indicated that the most populated site was near the NF-κB -DNA interface region. Thus, after the initial blind docking procedure, the 69 phenolic structures were docked onto the interface region of the NF-κB-DNA (Figure 5).

These docking simulations gave, as a result, 69 complexes with score values between −10.78 and −7.60 kcal/mol (Figure 5 and Figure 6).

A very interesting structural scenario emerges from the observation of the 50% best docking poses (Figure 5). On one hand, Site finder (Figure 5a) evidenced a cavity in the interface DNA-protein with polar (red spheres) and hydrophobic (grey sphere) properties. In Figure 5b, after a clustering procedure, we could visualize three preferred regions for the molecules (colored in green, red, or cyan). The green molecules are posed in the main groove of DNA, the cyan ones in the minor groove, and the red ones interact with a region more exposed to the DNA but not in the interface protein-DNA. This observation triggered the hypothesis that the identified structural diversity makes possible an ample range of strong interactions of phenols in the interface of the transcription factor NF-κB with DNA. Figure 5c reinforces and makes more evident the comprehension of such a structural scenario, and Figure 5d gives details about the residues that interact with quercetin, selected as the best docked phenol, as an example, the molecular environment of docked phenols.

Figure 6 provides a graphical representation of a very interesting structural fact characterizing the compounds mixture,: for the four identified scaffolds, an ample score range is evidenced for pigment compounds, with high scores for the flavan-3-ols, the phenolic acids, and the flavonols.

All classes display compounds with scores over the −9.19 kcal/mol, that is in the 50% percentile of higher ones. Then, when the “top five” are considered (Table 2), flavonols (the quercetin is the best ranked), flavan-3-ols in two polymerization stages (CCE4846 and CEE4846), (+)-catechin, and one anthocyan (Malvidin-3-*O*-(6-p-caffeoyl)-glucoside) are highlighted.

All of them have their own importance as bioactive compounds: quercetin is a key molecule in the neuroprotection hypothesis, anthocyanin subgroup, and are interestingly associated with the development of color in grapes and flavan-3-ols are considered as very interesting antioxidant and anti-inflammatory compounds. Increased longevity and a wide range of health benefits, including protection against cardiovascular disease and the progression of neurocerebral diseases, can be associated with flavonoid-rich intakes [29,30].

Throughout a plethora of studies, the view has emerged that the neuroprotective, anti-inflammatory and anti-cancer effects of flavonoids are probably mediated by interaction with specific proteins involved in intracellular signaling cascades. Currently, quercetin and related flavonoids are well-known inhibitors of phosphatidylinositol 3-kinase (PI3-kinase), xanthine oxidase [31], and cyclooxygenase [32], but beyond these targets, a complete understanding of the underlying mechanism of action of quercetin required the description of its entire target space [33]. A multimodal beneficial effect of quercetin has been associated with a multimodal mechanism, which would support the notion that quercetin has a major therapeutic potential [34].

Finally, additional considerations must be made regarding the relative composition of the main phenolic acid classes. From this viewpoint, it could be conjectured that in this so diverse mixture of compounds, flavonols and flavan-3-ols should have the biggest impact in terms of biological activity, with anthocyanins being less important given their relatively low content in the compounds mixture [35,36].

## 3. Materials and Methods

### 3.1. Chemicals and Instruments

Unless otherwise indicated, all chemicals used were of the highest available grade and purchased from Sigma Aldrich (St. Louis, MI, USA), Merck (Rahway, NJ, USA). Culture media, fetal bovine serum (FBS), and consumables for cell culture were obtained from Gibco, Thermo Fischer Scientific (Waltham, MA, USA). Plasticware was obtained from Corning (Corning, NY, USA).

Thermoregulated bath Branson 2510; UV-6300PC Double Beam Spectrophotometer, spectrometer Genesys 5, and EPR spectrometer Bruker ECS106 with a rectangular cavity and 50-kHz field modulation were used.

### 3.2. Grape Pomace Collection and Phenol Extraction 

Five grape pomace samples of Tannat and one of Cabernet Franc from Uruguay vineyards were collected in 2016. Frozen grape pomace samples were submitted to vacuum drying at 60 °C until constant weight. Dried grape pomace samples were ground and stored in darkness until the extraction was performed.

Maceration method was elected for the extraction of phenols. Briefly, 1.0 g of the dry powder was exposed to 25 mL of different ethanol/water mixtures (100/0, 80/20, 50/50, 20/80, and 0/100) for 7 days at room temperature protected from light. The extracts were vacuum-filtered, and the total phenolic content was determined for each extraction.

### 3.3. Antioxidant Capacity Assays

For these assays, 5.0 g and 10.0 g of grape pomace powder were extracted using 50/50 ethanol:water. The extracts were further characterized by Folin-Ciocalteau, DPPH, ABTS^·+^, FRAP, ORAC, and EPR.

#### 3.3.1. Total Phenolic Content Quantification

Total phenolic content was determined using the Folin–Ciocalteu (FC) method as described by Singleton and others [37]. The calibration curve was made using gallic acid, and the results were expressed in mg of gallic acid equivalents per liter (mgGAE/L).

#### 3.3.2. DPPH Assay

This colorimetric method is based on the radical DPPH [38]. Different extract solutions with 6.25, 12.5, 25, 37.5, and 50 g/mL concentrations of ascorbic acid were prepared, respectively. A total of 10 μL of each extract solution and 990 μL of DPPH^·^ (0.05 mM) were mixed, and the absorbance at 517 nm was measured after a 30 min incubation period in darkness at 30 °C. For the blank, DPPH was replaced with methanol. Additionally, a mixture of 10 μL methanol and 990 μL DPPH (0.05 mM) control solution was prepared. An ascorbic acid calibration curve was performed, and the results were expressed as mg ascorbic acid/L (mgAAS/L).

#### 3.3.3. ABTS Assay

The assay was performed according to Re et al. [39], with minor modifications [38]. The sample solutions were added to the resulting blue-green ABTS radical solution. The mixture was incubated in a water bath at 37 °C protected from light for 10 min. The decrease in absorbance at 734 nm was measured at the endpoint of 10 min. Methanol 50% and ABTS radical solution were used as a control. An ascorbic acid calibration curve was used, and the results were expressed as mg ascorbic acid/L.

#### 3.3.4. Oxygen Radical Antioxidant Capacity-Fluorescein (ORAC-FL)

This assay was performed, with minimal modifications, as described elsewhere [40]. Briefly, the standard protocol consists of quantitative measures of the fluorescein bleaching caused by the action of free radicals, using 2,2′-azobis(2-amidinopropane) dihydrochloride (ABAP) as the peroxyl radical generator.

Analyses were carried out in a Synergy HT Multi-Detection Microplate Reader (BioTek Instruments, Winooski, VT, USA) using polystyrene 96-well plate (Nunc, Denmark). Fluorescence was measured at an excitation wavelength of 485/20 nm and emission at 528/20 nm. The reaction was performed at 37 °C in phosphate-buffered solution (pH = 7.4) at a final volume of 200 μL. Fluorescein was prepared in buffer solution (40 nM); stock compound solutions were prepared in 50:50 ethanol:water and diluted in PBS to obtain working solutions. The mixture was pre-incubated for 15 min at 37 °C before the addition of the ABAP solution (final concentration, 18 mM). The fluorescence was recorded every 1 min for 120 min. A control assay with fluorescein, ABAP, and PBS was performed for each assay. Trolox was employed as a standard antioxidant at a final concentration range between 3 and 20 μM. The inhibition capacity was expressed as mM Trolox equivalent (TE) and was quantified by the integration of the area under the curve (AUC). The net AUC of each sample was calculated by subtracting the AUC corresponding to the control. Data processing was performed using Origin Pro 8 (Origin Lab Corporation, Northampton, MA, USA).

#### 3.3.5. Evaluation of the Ferric Reducing Antioxidant Power (FRAP Assay)

FRAP assay was performed based on the method of Benzie and Strain [41] using an automated plate reader set at 593 nm. An aliquot of 150 μL of sample solution were mixed in an amber flask with ultrapure water (450 μL), then with FRAP reagent (950 μL). The solution was stirred, and, subsequently, the absorbance was measured at room temperature. Stock solutions were prepared in ethanol and diluted with a phosphate buffer to afford sample concentrations between 15 and 58 μM ranges. FRAP values were expressed as μg of Trolox equivalents per mL (μg TE/mL).

### 3.4. Hydroxyl Radical Scavenging—Electron Spin Resonance

The Electron Spin Resonance (ESR) technique enables the detection and identification of paramagnetic species in different matrices and has been applied in several fields, such as food antioxidants [42,43]. Sample reactivity against hydroxyl radical was investigated using the non-catalytic Fenton type method [44]. Samples were prepared as follows: 100 μL of distilled water and 50 μL of NaOH (25 mM) were mixed, followed by the addition of 50 μL of DMPO spin trap (30 mM final concentration), sample 50 μL (1:1:2 in EtOH/H2O/DMF), and finally, 50 μL of hydrogen peroxide (30%). The mixture was put into an ESR cell, and the spectrum was recorded after five minutes of reaction. ESR spectra were recorded in the X band (9.85 GHz) using a Bruker ECS 106 spectrometer with a rectangular cavity and 50 kHz field modulation. The hyperfine splitting constants were estimated to be accurate within 0.05 G. The Scavenging activity of each sample was estimated by comparing the DMPO-OH adduct signals in the antioxidant-radical reaction mixture and the control reaction at the same reaction time, and it was expressed as the percentage of scavenging of hydroxyl radical.

### 3.5. HPLC-MS

The separation of phenol compound was performed using a Shimadzu 8040 triple quadrupole HPLC-DAD-MS/MS (Shimadzu, Tokyo, Japan) with a Kinetex C18-EVO reverse phase C18 column (5 μm particle size, 150 mm × 4.6 mm i.d.) (Phenomenex, Torrance, CA, USA) thermostated at 35 °C. Solvents used: (A) aqueous solution of trifluoroacetic acid (TFA) 0.1% and (B) 100% HPLC-grade acetonitrile. The mobile phase consisted of a six steps gradient of solvent B:1—1–40 min from 4% to 30%2—40–45 min from 30 to 98%3—45–47 isocratic regime of 98%4—47–48 from 98 to 5% and5—48–52 isocratic 5%.

Flow rate 1.3 mL/min, injection of 5 μL of samples diluted 1/4.

UV-Vis detection was carried out at 280 and 520 nm as the preferred wavelength. LC-MS analyses were performed using electrospray ionization (ESI) interface, source voltage was 2.50 kV, and capillary temperature was 250 °C. Mass spectra were recorded in positive ion mode between m/z 100 and 2000.

### 3.6. Cell Culture

HT-29-NF-κB-hrGFP cells [27] were cultured in RPMI 1640 supplemented with 10% (*v*/*v*) FBS in 25 cm^2^ tissue culture flasks. Cells were routinely cultured at 37 °C, 5% CO_2_ in a humidified incubator and subcultured when reaching approximately 70% confluence. Subsequently, cells were trypsinized, the concentration was adjusted, and the cells were seeded for different experimental settings. Cells with no more than 12 passages were used in all described assays.

### 3.7. NF-κB Activation Studies In Vitro

HT-29-NF-κB-hrGFP cells were seeded in 96-well plates in RPMI 1640 supplemented with 10% (*v*/*v*) FBS with a seeding density of 2 × 10^4^ cells/well and cultured overnight. After 24 h, the medium was renewed, and a 1/200 dilution of the different extracts was added simultaneously with the pro-inflammatory stimulus: TNF-α (10 ng/mL). Conditions were assayed in triplicate. Cells were further incubated for 24 h and, subsequently, trypsinized and resuspended for flow cytometry analysis. Cells were analyzed using a BD Accuri™ C6 (BD Bioscience, San Jose, CA, USA) flow cytometer equipped with 488 nm and 640 nm lasers. BD Accuri™ C6 software was used for data acquisition, and FlowJo v.10 for data analysis. Green Fluorescent Protein (GFP) and propidium iodide fluorescence emission was detected using band-pass filters 533/30 and 585/40, respectively. For each sample, 10,000 single events gated on a forward scatter (FSC) versus side scatter (SSC) dot plot were recorded. Only single living cells (those that excluded propidium iodide) were considered for results comparison. Cells without treatment and cells treated only with the stimulus or the different extracts were included as controls. NF-κB activation was normalized considering the TNF-α control as 100% NF-κB activation.

### 3.8. IL-8 Quantification in HT-29-NF-κB-hrGFP Culture Supernatants

Interleukin-8 (IL-8) levels were determined in cell culture supernatants using an ELISA kit according to the manufacturer instructions (ELISA MAX™ Deluxe Set, Biolegend, San Diego, CA, USA). HT-29-NF-κB-hrGFP cells were seeded in 96-well plates in RPMI 1640 supplemented with 10% (*v*/*v*) FBS with a seeding density of 2 × 10^4^ cells/well and cultured overnight. Medium was renewed, and a 1/200 dilution of the different extracts were added simultaneously with the pro-inflammatory stimulus: TNF-α (10 ng/mL) and cells were further incubated for 24 h. Culture supernatants were collected, clarified by centrifugation and stored at −20 °C until analysis.

### 3.9. Statistical Analysis

Depicted on graphs are the mean ±SD values of triplicates of a single experiment, and at least three independent experiments were executed. GraphPad Prism Software version 6.00 was used for statistical calculations. Student’s *t*-test was used when comparing two groups, and One-Way ANOVA test with Dunnett’s post-test was used to evaluate multi-comparison differences with a significance level of 95% (* *p* < 0.05).

### 3.10. Molecular Docking Simulations

Phenolic components were docked against the 2.90 Å crystallographic structure of the mouse NF-kB p50/p65 heterodimer complexed to the immunoglobulin kB DNA from Protein Data Bank (PDB: 1VKX).

As a first step, a blind docking was performed using a database containing 69 phenols previously found in the HPLC-MS analysis as putative ligands. The blind docking was performed using Autodock Vina, and the entire receptor structure was included in the search space. After analyzing the results of blind docking, the most populated site was identified near the NF-kB-DNA interface region, localized in the DNA major groove. A Site Finder search confirmed the interface between NF-kB and the DNA as the best site to use for further docking studies. Then, a second molecular docking procedure with the same phenolic database was carried out in this specific interface region. The Affinity Scoring function, implemented in Autodock Vina, was used, and the best nine poses for each molecule were retained.

## 4. Conclusions

In this work, we have described the characterization of a set of grape pomace samples from Uruguayan vineyards in terms of total phenolic content, antioxidant capacity, and anti-inflammatory activity. The latter was assessed through NF-κB modulation and IL-8 production.

HPLC-MS experiments allowed the unambiguous identification of 43 compounds with the large prevalence of flavan-3-ols, followed by flavonols, phenolic acids, and anthocyanins. Furthermore, the extracts displayed the ability to modulate NF-κB activity in a model of epithelial intestinal reporter cells. For this reason, the identified compounds were further characterized in silico through docking simulations on the NF-κB -DNA complex as a target.

Docking scores ranged between −7.6 and −10.78 kcal/mol, confirming their potential ability to bind the target. In addition, all classes of molecules were characterized by compounds with scores in the high 50% percentile (binding affinity higher than −9.19 kcal/mol). The “top five” flavonols were the quercetin aglicone (the best ranked), flavan-3-ols in two polymerization stages (CCE4846 and CEE4846), (+)-catechin, and one anthocyan (Malvidin-3-*O*-(6-p-caffeoyl)-glucoside).

When the average percentage measured in the HPLC-MS experiment is taken into account, it can be concluded that in this so diverse mixture, the presence of flavonols and flavan-3-ols should have the greatest impact on the biological activity of the extracts. On the other hand, anthocyans, despite their relatively low presence in the mixture, should give a significant contribution to the biological activity given their putative strong interaction with the NF-κB -DNA complex evidenced by docking simulations.

Even among the samples from the same grapes variety, a wide range of total phenolic content and different antioxidant activity were detected, with Tannat B being the extract with the highest antioxidant and anti-inflammatory activity *in vitro*.

Finally, as grape pomace is a winery byproduct with low commercial value, results reported in this work indicate that the tannat pomace extracts are the best source of grape pomace antioxidants and low-cost raw material that can be processed by the nutraceutical or functional foods industry to obtain products with high added value and interesting commercial value.

## Figures and Tables

**Figure 1 molecules-28-03909-f001:**
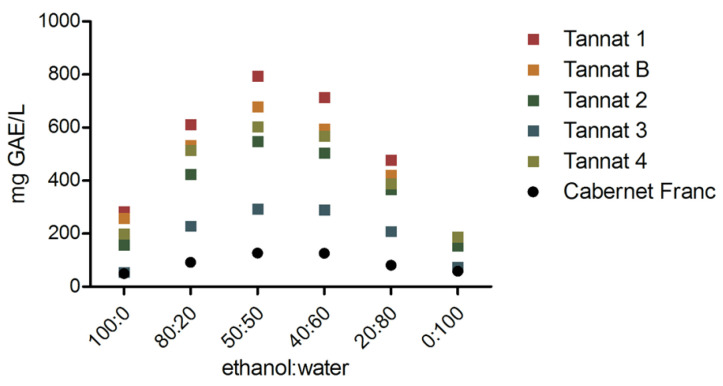
Amount of polyphenols measured by the Folin–Ciocalteu (FC) method, expressed as mg GAE/L, plotted against the solvent extraction composition.

**Figure 2 molecules-28-03909-f002:**
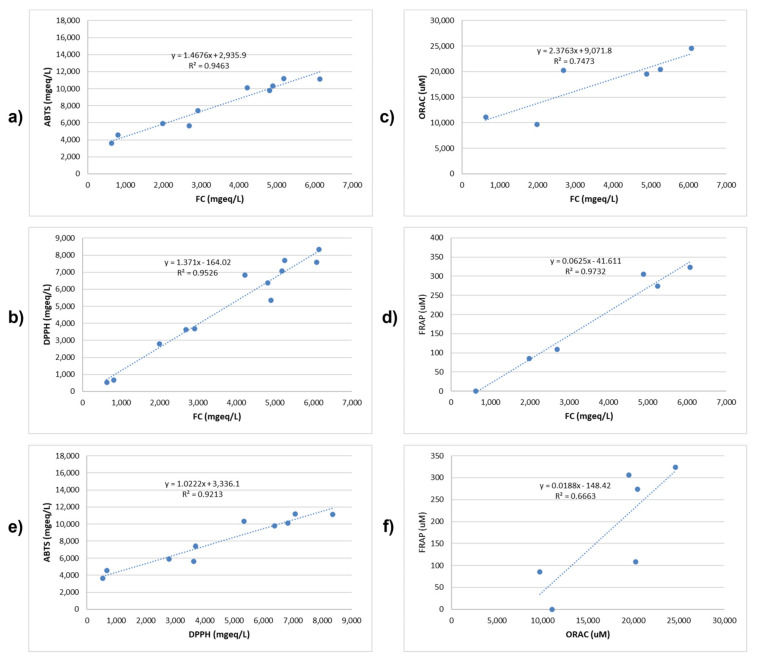
(**a**–**d**): Physicochemical measurement and relationships of global phenol contents (Folin Ciolcateu) with antioxidant capacity of bioactive phenols contained in the extraction from 10 g of dry grape pomace samples under study measured as ABTS, DPPH, ORAC, and FRAP. (**e**,**f**): Relationships between ABTS and DPPH and between ORAC and FRAP rates. GAE: gallic acid equivalents; AAS: ascorbic acid; TE: Trolox Equivalent. Units used in each determination are Folin Coilcateu in mgGAE/L, ABTS and DPPH in mgeq AAS/L, ORAC in (mM TE), and FRAP in ugTE/mL.

**Figure 3 molecules-28-03909-f003:**
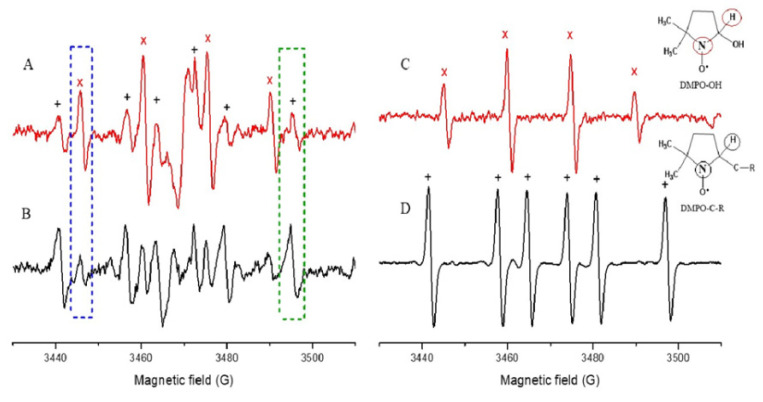
ESR spectra corresponding to (**A**) Generation of radicals in the solvent mixture; (**B**) Decrease in the signals resulting from the addition of the tannat extract B; (**C**) Characteristic signals of the DMPO-OH spin adduct (x); and (**D**). characteristic signals DMPO-CR (+). In blue, the decrease in the intensity of the DMPO-OH adduct signal, and in green, the increase in the DMPO-CR intensity is shown.

**Figure 4 molecules-28-03909-f004:**
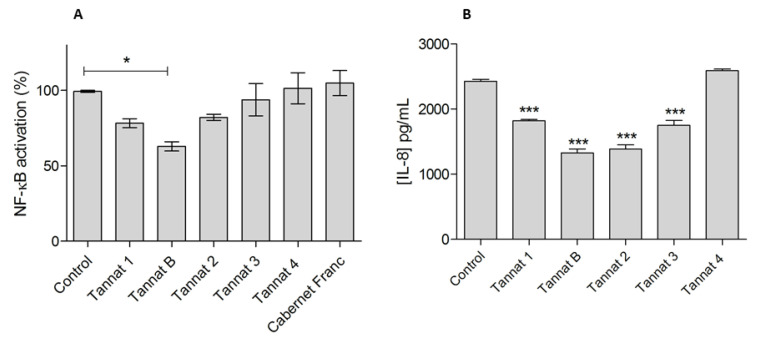
Anti-inflammatory activity of grape pomace. Modulation of TNF-α-induced NF-κB signaling in HT-29-NF-κB-hrGFP (**A**) and IL-8 determination in supernatants (**B**). Cells were treated with different red grape pomace extracts and 10 ng/mL of TNF-α for 24 h. NF-κB activation and cell viability were evaluated by flow cytometry. In all cases, cell viability was over 90%. Depicted are the Mean ± SD values of triplicates of one representative experiment. TNF-α control was used for One-way ANOVA analysis (Dunnett’s post-test) to compare with the treated groups, * *p* < 0.005; *** *p* < 0.0001.

**Figure 5 molecules-28-03909-f005:**
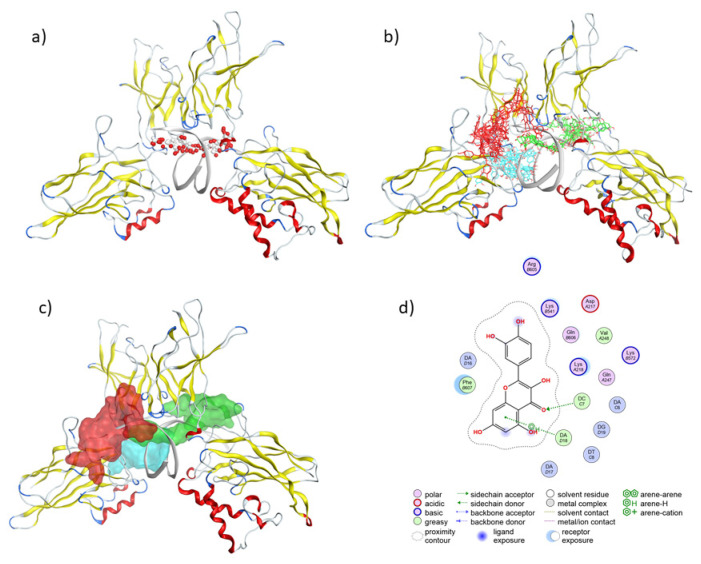
(**a**) Biggest Site identified by Site Finder, drawn as grey (apolar) and red (polar) alpha spheres, placed in the interface between the major groove of DNA; (**b**) The 50% of best docked conformations are shown in cyan (DNA minor groove) and green (DNA major groove and red (a site outside the DNA-protein interface; (**c**) Best docked site, placed in the interface between the major groove of DNA and the transcription factor NF-κB drawn as a green molecular site surrounding the docked phenols; (**d**) The best scoring phenol, quercetin, is shown together with the main contacts, in a two-dimensional view. Color coding of the spheres is given in the legend.

**Figure 6 molecules-28-03909-f006:**
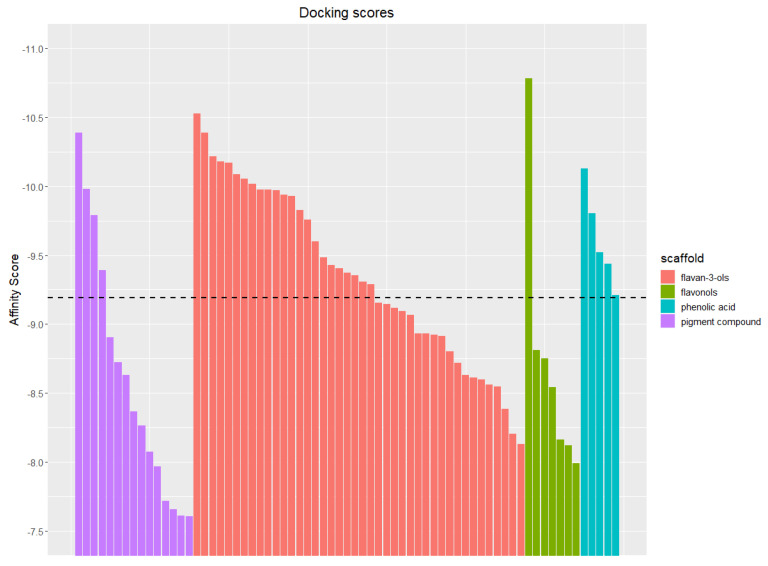
Affinity Score (kcal/mol) of the full set of studied phenols, grouped in four scaffolds: flavan-3-ols (orange), flavonols (green), phenolic acids (cyan), and pigment compounds (anthocyananthocyans, purple). A dotted black line marks a score level of 50% (−9.19 kcal/mol) of total range (−7.60 to −10.78 kcal/mol).

**Table 1 molecules-28-03909-t001:** Identified phenolic compounds from the most bioactive grape extract. In the second and third columns, the identified name and the structural class are annotated. Fourth column: λ_max_: wavelength (expressed in nanometers) in which the HPLC signal is registered; fifth column: Rt: retention time expressed in minutes; AUC: is the area under the curve measured in absorbance units (mA). Last column: percentage of each identified compound.

ID	Name	Structural Class	λ_max_	Rt	AUC	Fraction (%)
1	delphinidin-3-*O*-glucoside	anthocyan	520	16.377	44,475	0.148
2	cyanidin-3-*O*-glucoside	anthocyan	520	18.349	8867	0.029
3	petunidin-3-*O*-glucoside	anthocyan	520	19.396	74,001	0.246
4	peonidin-3-*O*-glucoside	anthocyan	520	21.361	30,264	0.101
5	malvidin-3-*O*-glucoside	anthocyan	520	22.12	246,379	0.818
6	delphinidin-3-*O*-(6′-acetyl)glucoside	anthocyan	520	23.821	5490	0.018
7	petunidin-3-*O*-(6′-acetyl)glucoside	anthocyan	520	26.6	15,340	0.051
8	peonidin-3-*O*-(6′-acetyl)glucoside	anthocyan	520	29.129	9585	0.032
9	malvidin-3-*O*-(6′-acetyl)glucoside	anthocyan	520	29.361	47,261	0.157
10	delphinidin-3-*O*-(6′-p-coumaroyl)glucoside	anthocyan	520	29.799	52,391	0.174
11	malvidin-3-*O*-(6′-caffeoyl)glucoside	anthocyan	520	31.465	21,549	0.072
12	cianidin-3-*O*-(6′-p-coumaroyl)glucoside	anthocyan	520	31.803	16,157	0.054
13	petunidin-3-*O*-(6′-p-coumaroyl)glucoside	anthocyan	520	32.293	81,375	0.270%
14	peonidin-3-*O*-(6′-p-coumaroyl)glucoside	anthocyan	520	34.49	70,515	0.234
15	malvidin-3-*O*-(6′-p-coumaroyl)glucoside	anthocyan	520	34.68	467,693	1.553
	**total pigment compounds**					**3.957**
16	cis-caftaric acid	phenolic acid	280	3.806	68,021	1.647
17	trans-caftaric acid	phenolic acid	280	5.072	12,280	0.297
18	trans-cutaric acid	phenolic acid	280	7.292	173,386	4.197
19	p-coumaroyl hexose (1)	phenolic acid	280	9.805	42,092	1.019
20	p-coumaroyl hexose (2)	phenolic acid	280	11.058	80,846	1.957
	**total phenolic acid**					**9.117**
21	procyanidin trimer C2	flavan-3-ols	280	5.8	13,376	0.324
22	procyanidin dimer B1	flavan-3-ols	280	10.223	53,601	1.297
23	procyanidin dimer B3	flavan-3-ols	280	12.037	73,849	1.788
24	(+)-catechin	flavan-3-ols	280	12.547	289,603	7.010
25	procyanidin trimer	flavan-3-ols	280	13.581	176,894	4.282
26	procyanidin trimer	flavan-3-ols	280	14.213	193,870	4.693
27	procyanidin dimer B4	flavan-3-ols	280	15.154	771,559	18.677
28	procyanidin dimer B6	flavan-3-ols	280	15.687	135,473	3.279
29	(-)-epicatechin	flavan-3-ols	280	16.39	335,850	8.130
30	procyanidin dimer galloylated	flavan-3-ols	280	17.778	13,224	0.320
31	procyanidin trimer	flavan-3-ols	280	18.119	132,934	3.218
32	procyanidin trimer	flavan-3-ols	280	18.784	168,691	4.083
33	procyanidin dimer B2	flavan-3-ols	280	19.494	132,598	3.210
34	procyanidin dimer galloylated	flavan-3-ols	280	19.821	313,241	7.582
35	(-)-epictechin gallate	flavan-3-ols	280	23.44	275,773	6.676
36	procyanidin dimer B7	flavan-3-ols	280	24.65	31,102	0.753
	**total flavan-3-ols**					**75.322**
37	myricetin-3-*O*-galactoside	flavonols	280	20.323	88,291	2.137
38	myricetin-3-*O*-glucoside	flavonols	280	21.371	93,796	2.270
39	quercetin-3-*O*-galactoside	flavonols	280	24.006	42,083	1.019
40	quercetin-3-*O*-glucoside	flavonols	280	24.659	94,150	2.279
41	siringetin-3-*O*-glucoside	flavonols	280	28.014	8886	0.215
42	quercetin-7-*O*-neohesperidoside	flavonols	280	29.795	65,789	1.593
43	quercetin aglycone	flavonols	280	36.718	86,379	2.091
	**total flavonols**					**11.604**
	**total compounds identified**					**100.000**

**Table 2 molecules-28-03909-t002:** The five compounds with the best score were identified through the second molecular docking simulations. Last column: Affinity Score is in kcal/mol. In parentheses, % indicates the percentage of each compound class with respect to the total number of compounds, and Av% is this percentage divided by the number of compounds in that class.

ID	Molecule	Type	Affinity (%/Av %)
1	Quercetin aglycone	Flavanol	−10.78 (11.6/1.7)
2	(+)-catechin	Flavan-3-ols	−10.53 (75.32/4.7)
3	CCE4846	Flavan-3-ols	−10.39 (75.32/4.7)
4	Malvidin-3-*O*-(6-p-caffeoyl)-glucoside	Anthocyan	−10.39 (3.96/0.3)
5	CEE4846	Flavan-3-ols	−10.22 (75.32/4.7)

## Data Availability

All data are available upon request to the authors.

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
