# Peer review of "Antioxidant Capacity and NF-kB-Mediated Anti-Inflammatory Activity of Six Red Uruguayan Grape Pomaces"

_molecules, 2023, doi:10.3390/molecules28093909_

Round 1
Reviewer 1 Report
The work is focused on the study of the valorisation of different red uruguayan grape pomaces in order to have extracts with high antioxidant and anti-inflammatory potential. To this purpose, the optimization of the extraction in terms of total polyphenols by using different solvent proportions was carried out and then, the extracts were assessed for the antioxidant activity by different methods and for the in vitro anti-inflammatory activity. This work contributes to the study of natural sources for bioactive compounds, so I recommend it to be published but, first, minor corrections should be made:
Introduction:
- Line 45: please, give de reference for the last statement.
- Line 57: could be given data about the production of grape pomace as the solid waste from wine industries?
- Line 60: it would be advisable to describe the most common polyphenols found in the grape pomace extractives.
- In this section, a review of the background over the years of grape pomace extraction for the obtention of bioactivity phenols is missing.
Results and Discussion:
- Please, review the subsections numeration. Number 2.5 is repeated.
- Line 77-79: delete.
- Line 120: please review the figure 2 description title.
- In line 111, it is said that 5g samples are shown in figure 2, but in the figure 2 title is said to be 10g samples, please check.
- Figure 2: add the name and the unit for the axes.
- Line 163: it should be said “Figure 2c”, check.
- Line 168: review if “FC method” is correct here or if it should be “FRAP method”
- Lin1 182: number correctly the figure, is Figure 3.
- Line 188: add these data to the Supplementary file.
- Line 203: delete “9”.
- Table 1: review the title information: the identified name and the structural class are in second and third columns, also express the AUC information/units.
- Line 215: Also mention the phenolic acids class.
- Line 222: place the brackets properly.
- Line 235: review the spelling.
- Figure 4: explain the level of significance for (*).
- Lines 323 and 327: renumber the table, is Table 2.
- Line 331-334: please, give the references.
- Line 333: correct the spelling of “flavan-e-ols”
- A general review and comparison of the obtained data with other in bibliography is missing: for example, the profile of phenols obtained from grape pomace by other authors and its implications on biological activity.
Materials and Methods:
- Line 359: explain why 5.0 and 10.0 grams were used here if in the grape pomace extraction is said that 1g was used.
Conclusions:
- Line 497: check the spelling.
- Line 501: “10,78 kcal/mol” is a negative value, the negative symbol should be places also before the number.
- Line 504: replace “.” by “,”.

Author Response
We would like to thank the Reviewer 2 for the thorough revision of our manuscript and comments that have undoubtedly contributed to improve the quality of our work and the scope of our research.
Below, we provide a point-by-point answer to the comments and remarks (indicated as AR, authors’ responses). Changes to the manuscript text are highlighted in yellow in the marked manuscript file. Please note that the lines numbering to which the Reviewer is referring is different from that of the revised version of the manuscript.
Introduction:
- Line 45: please, give de reference for the last statement.
AR: A new reference was included. Pizzino et al., Oxid Med Cell Longev. 2017; 2017: 8416763. doi: 10.1155/2017/8416763
- Line 57: could be given data about the production of grape pomace as the solid waste from wine industries?
AR: We have included 2022 official data from the Instituto Nacional de Vitivinicultura (INAVI), Uruguay (https://www.inavi.com.uy/estadisticas/).
- Line 60: it would be advisable to describe the most common polyphenols found in the grape pomace extractives.
AR: A new reference was included: García-Marino et al., Analytica Chimica Acta, 563 (2006) 44-50.
- In this section, a review of the background over the years of grape pomace extraction for the obtention of bioactivity phenols is missing.
AR: Reviews of the background over the years have been included and new references were introduced.
Quideau et al., Angew. Chem. Int. Ed. 2011, 50, 586 – 621
Castellanos-Gallo et al., Processes 2022, 10(3), 469; https://doi.org/10.3390/pr10030469.
Results and Discussion:
- Please, review the subsections numeration. Number 2.5 is repeated.
AR: Subsections numeration and repetitions were revised and corrected.
- Line 77-79: delete.
AR: They were deleted.
- Line 120: please review the figure 2 description title.
AR: Figure 2 description title and caption were revised and rewritten.
- In line 111, it is said that 5g samples are shown in figure 2, but in the figure 2 title is said to be 10g samples, please check.
AR: We have checked and reworded this paragraph in the revised manuscript’s version as followed
In this work, different methods associated with different antioxidants mechanisms were used and the results for 10 g weight samples are shown in Figure 2 (see Suplementary Table S1 for the results from 5 and 10 g). Since the total phenolic contents and the antioxidant capacity were similar between 5 and 10 g sample dry weight, the extraction obtained from 10 g dry weight was used for further analysis.
- Figure 2: add the name and the unit for the axes.
AR: Names and unit of measures for the axes were added.
- Line 163: it should be said “Figure 2c”, check.
AR: It was checked and it is correct.
- Line 168: review if “FC method” is correct here or if it should be “FRAP method”
AR: we have replaced the wrong term “FC method” with the correct one, “FRAP method”.
- Lin1 182: number correctly the figure, is Figure 3.
AR: We apologize for this oversight, in the revised version is correctly numbered as Figure 3
- Line 188: add these data to the Supplementary file.
AR: Instead of including the data of the decreased signal intensity of spin adduct DMPO-OH as Supplementary file we have decided to provide them under request. Therefore, the text was changed as follows:
A decreased signal intensity of spin adduct DMPO-OH was observed for all tested samples (data not shown, available upon request to the authors).
- Line 203: delete “9”.
AR: Number 9 was deleted.
- Table 1: review the title information: the identified name and the structural class are in second and third columns, also express the AUC information/units.
AR: The title information was reviewed and the meaning of AUC was included in the heading of the table as followed:
AUC: is the area under the curve measured in absorbance units (mA).
- Line 215: Also mention the phenolic acids class.
AR: The phenolic acids class is now mentioned in the revised version.
- Line 222: place the brackets properly.
AR: Done
- Line 235: review the spelling.
AR: The spelling has been corrected
- Figure 4: explain the level of significance for (*).
AR: The significance level for (*) is p<0.05. This has now been included to the Figure 4 caption.
- Lines 323 and 327: renumber the table, is Table 2.
AR: The Table 2 is now correctly numbered.
- Line 331-334: please, give the references.
AR: References were included at the end of the paragraph (Commenges et al., 2000; McCullough et al., 2012)
- Line 333: correct the spelling of “flavan-e-ols”
AR: The spelling was corrected by of flavan-3-ols.
- A general review and comparison of the obtained data with other in bibliography is missing: for example, the profile of phenols obtained from grape pomace by other authors and its implications on biological activity.
AR: A general review and comparison of the obtained data with other in bibliography was made and XXX more references were included:
Peixoto et al., Food Chemistry, 2018, 253, 132-138. https://doi.org/10.1016/j.foodchem.2018.01.163
Onache et al., Separations 2022, 9(12), 395; https://doi.org/10.3390/separations9120395
Pistol, G.C et al. Br. J. Nutr. 2019, 121, 291–305.
Calabriso, N et al. Nutrients 2022, 14, 1175.
Materials and Methods:
- Line 359: explain why 5.0 and 10.0 grams were used here if in the grape pomace extraction is said that 1g was used.
AR: Thank you for pointing this out. In the Section 2.1 an explanation about this concern was included (see the paragraph below). Furthermore, once the extraction method was optimized (using 1 g dry weight and 50:50 ETOH:Water mix), the antioxidant capacity was evaluated for the six samples of grape pomace using two sample weights (5 g and 10 g).
In order to select the best mixture that yields high polyphenols content, different solvent mixtures and sample weights were employed. On a first step and aiming to find the ethanol/water relation that extracted the most polyphenols, 1g of dry pomace samples were submitted to maceration and extraction with different ethanol/water mixtures (100/0, 80/20, 40/60, 50/50, 20/80 and 0/100). Results are shown in Figure 1, where the total phenolic content is expressed in mg GAE/L.”
Conclusions:
AR: Conclusions were rewritten
- Line 497: check the spelling.
AR: The misspelling was eliminated
- Line 501: “10,78 kcal/mol” is a negative value, the negative symbol should be places also before the number.
AR: It was corrected, it is a negative number
- Line 504: replace “.” by “,”.
AR: Done
Reviewer 2 Report
The study related to antioxidant and anti-inflammatory properties of six Red Uruguayan grape pomaces was observed to have comprehensive chemical analyses and good numbers of experiment in a one way. On the other hand, the manuscript was seemed too one sided. There should be further biological analyses that could constitute a better approach.
1- Template information should be deleted from the manuscript like;
“This section may be divided by subheadings. It should provide a concise and precise description of the experimental results, their interpretation, as well as the experimental conclusions that can be drawn.”
“Please turn to the CRediT taxonomy for the term explanation.”
2- Language check should be performed.
3- Cytotoxicity analyses of the isolates on the health human cell line or blood culture should be performed to investigate potential usability.
4- Antioxidant and anti-inflammatory properties of the isolates were instigated in the manuscript. On the other hand, possible application of the extracts could be analysed in respect to hepatoprotection, neuroprotection, antiproliferation, antimicrobial or antiviral. There is a key point is lacking in the study.
5- Conclusion part is more of a summary of the results; this part should be modified in a conclusive way.
Author Response
We would like to thank the Reviewer 3 for the thorough revision of our manuscript and comments that have undoubtedly contributed to improve the quality of our work and the scope of our research.
Below, we provide a point-by-point answer to the comments and remarks (indicated as AR, authors’ responses). Changes to the manuscript text are highlighted in yellow in the marked manuscript file. Please note that the lines numbering to which the Reviewer is referring is different from that of the revised version of the manuscript.
1- Template information should be deleted from the manuscript like;
AR: We have removed the template information.
2- Language check should be performed.
AR: We have performed a careful language check and corrected errors and typos.
3- Cytotoxicity analyses of the isolates on the health human cell line or blood culture should be performed to investigate potential usability.
AR: In our lab the cytotoxicity of different grape pomace extracts on health human cell lines, such HEK 293 (CRL-1573™ from ATCC, human embryonic kidney cells) and HaCat-SF (CLS 300493, immortalized human keratinocytes) were evaluated. In all cases, regardless grape pomace extracts or cell line, the IC50 values were in the range of 150 to 300 mg GAE/L. Particularly, in the present study, all the extracts were used at least 5 to 10 times fold-diluted, ensuring their potential usability.
Furthermore in the biological assays performed in this study (Modulation of TNF-α-induced NF-κB signaling in HT-29-NF-κB-hrGFP), the cell viability was always evaluated by flow cytometry. In all cases, cell viability was over 90%, demonstrating that the grape pomace extracts were non toxic at the used dilutions. See legend to Figure 4.
4- Antioxidant and anti-inflammatory properties of the isolates were instigated in the manuscript. On the other hand, possible application of the extracts could be analysed in respect to hepatoprotection, neuroprotection, antiproliferation, antimicrobial or antiviral. There is a key point is lacking in the study.
AR: We thank the reviewer for pointing these out and we believe that the raised issues regarding cytotoxicity analyses, hepatoprotection, neuroprotection, antiproliferation, antimicrobial or antiviral possible application of the extracts are highly relevant.
In the revised version of the manuscript, we have discussed these issues in the light of the data already available in the literature. Thus, several new references were included in the manuscript. On the other hand, we believe that carrying out the extensive experimental studies proposed by the Reviewer would require a substantial additional experimental effort that, in our opinion, is beyond the scope of the present paper. Indeed, part of the studies proposed by the Reviewer in combination with new studies that we are conducting will be the object of a follow up study that will produce data for a new paper.
5- Conclusion part is more of a summary of the results; this part should be modified in a conclusive way.
AR: We have modified the Conclusion section following the Reviewer's advice.
Round 2
Reviewer 2 Report
The manuscript could be accepted in the present form.